# Modified Magnesium Oxysulfate Foam Cement Doped with Iron Tailings

**DOI:** 10.3390/ma17235907

**Published:** 2024-12-02

**Authors:** Yitong Fang, Baoluo Xu, Lisha Fu, Le Chen, Zilong Chen, Wanjun Hao, Kexi Zhang

**Affiliations:** State Key Laboratory of South China Sea Marine Resources Utilization, School of Materials Science and Engineering, Hainan University, Haikou 570100, China; 18907670406@163.com (Y.F.); itspaul@163.com (B.X.); fls_0209@163.com (L.F.); 13407947390@163.com (L.C.); 15103087525@139.com (Z.C.)

**Keywords:** magnesium oxysulfate foam cement, iron tailing powder, compressive strength, water absorption, pore size distribution

## Abstract

The enhancement of the utilization rate of solid waste, along with balancing the comprehensive performance of materials, presents a significant challenge in the development of new functional building materials. This study examined the effects of high concentrations of iron tailing powder on the crystallization characteristics, pore structure, compressive strength, and water absorption of modified magnesium oxysulfate (MOS) foam cement with different dry densities. Furthermore, employing chemical foaming technology, the study characterized and analyzed the microstructure of modified MOS foam cement hydration products through scanning electron microscopy (SEM) and X-ray diffraction analysis (XRD). The results indicated that the addition of an acidic modifier effectively facilitated the hydration reaction in the MgO-MgSO_4_-H_2_O system, enhancing the micro-crystallization characteristics of MOS foam cement. The internal pores were uniformly round, with a dense crystal structure within the pore walls. The compressive strength of the material with 40% dry density A08 grade iron tailing powder reached 6.83 MPa, and the lowest water absorption was 5.32% at a dry density of A09.

## 1. Introduction

Currently, foam concrete is primarily a porous material prepared by mixing Portland cement with prefabricated foam. This material often faces issues, such as a low compressive strength and high water absorption [1,2,3,4,5,6,7]. This is attributable to the crystallization characteristics of the material and its internal pore structure. The crystallization of Portland cement-based foam concrete is suboptimal when a large amount of external reinforcement waste is present. This, combined with an unstable foam pore structure, leads to decreased material strength, increased susceptibility to cracking, high water absorption, and overall performance degradation [8,9,10,11,12]. Different porosities and pore sizes can affect the strength of concrete [13].

Magnesium oxysulfate cement (MOS) is a new type of gas-hard magnesia cementitious material. It is prepared by the reaction of light-burned magnesium oxide, magnesium sulfate heptahydrate, a foaming agent, and water. Furthermore, MOS offers advantages, such as a low energy consumption, high-temperature resistance, light weight, and high strength [14,15]. Additionally, it presents a cost-effective manufacturing process for foam concrete [16]. At present, the production methods of magnesium oxide include vapor phase method, acid dissolution method, calcination method, and precipitation method, among others [17]. Magnesium oxychloride foam cement can be used as thermal insulation and conventional energy-saving building material [18]. The accumulated stockpile of iron tailings has reached 5 billion tons, yet the utilization rate is less than 10% [19]. In Changjiang, Hainan Province, China, a large amount of solid-waste iron tailing powder is produced annually from the iron mines [20], resulting in serious environmental pollution and significant safety hazards [21,22,23]. Therefore, this study investigated the utilization of MOS as a base material, combined with iron tailing powder from Changjiang, Hainan, and employing chemical foaming technology to develop new light-weight and high-strength foam-concrete building materials [24,25,26,27,28,29,30]. This study aimed to explore the relationship between the internal crystallization characteristics, pore structure, mechanical properties, and water resistance of MOS foam cement containing large amounts of solid waste. The phase composition and microstructure of the experimental samples were analyzed through scanning electron microscopy (SEM) and X-ray diffraction (XRD). This study developed light-weight and high-strength MOS foam cement materials, achieving a low energy consumption and high-cost effectiveness in foam concrete application [31,32]. Thus, the study offered insights into the exploration of the functionalization and structural assembly of new building materials [33,34].

## 2. Materials and Methods

### 2.1. Materials

The light-burned magnesium oxide utilized in this study was produced by Yingkou Mingxin Magnesium Industry Company from Yingkou City, Liaoning Province, China, and it was characterized using the hydration method. The active magnesium oxide content was 66.93%, appearing as a white-to-light-yellow powder. The chemical composition and microstructure of the magnesium oxide are presented in Table 1 and Figure 1a. Industrial magnesium sulfate was sourced from Shandong Aochuang Chemical Co., Ltd., China. The effective content of MgSO_4_·7H_2_O was 99.06%, appearing as white or colorless soluble crystals. The iron tailing powder was obtained from the iron mines in Shilu Town, Changjiang Li Autonomous County, Hainan Province, China. Its primary components were 37.9% silicon dioxide and 38.7% iron oxide. The chemical composition and micro-morphology of the iron tailing powder are presented in Table 2 and Figure 1b. The composite modifier primarily consisted of organic acids such as citric acid. The foam stabilizer employed was chemically pure calcium stearate powder produced by Tianjin Damao Chemical Reagent Factory, China.

### 2.2. Preparation of Magnesium Oxysulfate Foam Cement

Initially, magnesium sulfate heptahydrate and water were measured in a set molar ratio (n(H_2_O): n(MgSO_4_)) of 20 to prepare a solution, and an acidic modifier was added to ensure complete dissolution. This solution was then placed in a cement paste mixer ac-cording to a molar ratio (n(MgO): n (H_2_O)) of 7. Light-burned magnesium oxide was added and stirred for 3 min–5 min to obtain MOS cement slurry. The foaming agent and foam stabilizer were mixed in different proportions and added to the slurry. After mixing with the MOS cement paste and stirring for ~45 s, the mixture was injected into molds. After curing for 24 h, the molds were de-molded, and the samples were placed at room temperature (20 °C ± 2 °C, relative humidity 65%). The samples were naturally cured for 28 d, after which their relevant properties were tested.

### 2.3. Characterization

The Chinese national standard JG/T 266-2011 was followed for processing. The dry density and water absorption rate were calculated. Cube specimens of 40 mm × 40 mm × 40 mm cement paste were made. After natural curing, the compressive strength test was conducted using a WDW-100 C microcomputer-controlled electronic universal testing machine, and the reaction mechanism was analyzed. The cross-section morphology was observed using a JSM-6390 A field emission scanning electron microscope manufactured by Japan Electronics Co., Ltd. A Smart Lab X-ray diffractometer produced by Rigaku Company in Japan was employed with a Cu target to control the scanning rate at 5°/min, and the scanning range was set from 10° to 80°. Image-Pro-Plus 6.0 software was used to analyze the cross-section morphology of the sample. After image acquisition and statistical processing, the pore structure was analyzed.

## 3. Results and Discussion

### 3.1. Effect of Iron Tailing Powder Content on Compressive Strength and Water Absorption

To evaluate the influence of iron tailing powder on the properties of MOS foam cement, the dry density was maintained at 800 (kg/m^3^). The mix ratio and basic mechanical properties of the cement, incorporating varying amounts of iron tailing powder, are detailed in Table 3. The powder content ranged from 10% to 60%, with the proportion of foam stabilizer and modifier expressed as a mass percentage of MgO. The effect of iron tailing powder content on compressive strength and water absorption is shown in Table 3.

Mix proportion: The molar ratio of MgO/MgSO_4_/H_2_O is 7:1:20 (The amount of composite modifier is 4‰).

The data in Table 3 indicate that at a dry density of 800 kg/m^3^, the MOS foam cement with 10% iron tailing powder exhibited a compressive strength of 4.45 MPa and a water absorption of 9.17%. Furthermore, as the iron tailing powder content increased, the compressive strength of the cement initially increased, peaking at 40%, where it achieved a maximum strength of 6.82 MPa. Subsequently, the performance declined sharply. Similarly, the water absorption rate decreased to a minimum of 7.21% at 40% powder content, indicating this as the optimal admixture proportion.

It was evident that incorporating an appropriate amount of iron tailing powder enhanced the mechanical properties and reduced the water absorption of the cement. The fine, uniformly distributed particles of iron tailings (1 μm–30 μm) effectively filled the MOS cementitious system [12], thus enhancing its initial density and compactness. Furthermore, optimizing the iron tailing powder content resulted in significantly stronger and more water-resistant MOS foam cement. However, the excessive addition of iron tailing powder disrupted the hydration reaction of the cement and the internal structure, increasing both porosity and water absorption, as well as reducing compressive strength.

Zhang Na et al. [13] explained the hydration process for modified sulfate-magnesia cement. Upon dissolution of MgSO_4_·7H_2_O in water, specific hydrated magnesium ions were formed within the solution. As a result of the polarization effect, specific [Mg(H_2_O)_6_]^2+^ ions underwent hydrolysis reactions (Equations (1) and (2)). Simultaneously, with the assistance of acidic modifiers, the hydrolysis rate of MgO accelerated and combined with water molecules on the surface of particles to form $[Mg(\mathrm{OH})(H_{2}O{)}_{X}{]}^{+}$ (Equation (3)). The acidic modifier hydrolyzes to produce CA^n−^ and H^+^. The OH^-^ ions in the system reacted with the H^+^ provided by the acidic modifier to form H_2_O (Equation (4)). The modifier anion CA^n-^ adsorbed $[Mg(\mathrm{OH})(H_{2}O{)}_{X}{]}^{+}$ to form a complex (Equation (5)) [35]. This complex reacted with $SO_{4}^{2-}$, OH^−^, Mg^2+^, and other particles in the system to form $5{\mathrm{Mg}(\mathrm{OH})}_{2}\cdot {\mathrm{MgSO}}_{4}\cdot 7H_{2}O$ (5·1·7 phase), as shown in Formula (6).
(1)$$\mathrm{MgS}O_{4}\cdot 7H_{2}O_{\left(s\right)}\to {\left[\mathrm{Mg}(H_{2}O{)}_{6}\right]}_{\left(\mathrm{aq}\right)}^{2+}+2SO_{4\left(\mathrm{aq}\right)}^{2-}$$
(2)$$[Mg(H_{2}O{)}_{6}{]}_{\left(\mathrm{aq}\right)}^{2+}\Leftrightarrow {\left[\begin{array}{c} Mg(OH)(H_{2}O{)}_{5} \end{array}\right]}_{\left(\mathrm{aq}\right)}^{+}+H_{\left(\mathrm{aq}\right)}^{+}$$
(3)$${\mathrm{MgO}}_{\left(s\right)}+(x+1)H_{2}O\to [\mathrm{Mg}(\mathrm{OH})(H_{2}O{)}_{\chi }{]}_{\left(\mathrm{surface}\right)}^{+}+OH^{-}$$
(4)$$OH^{-}+CA^{n-}+H^{+}=H_{2}O+CA^{n-}$$
(5)$$CA^{n-}+[\mathrm{Mg}(\mathrm{OH})(H_{2}O{)}_{\chi }{]}_{\left(\mathrm{surface}\right)}^{+}\to [\mathrm{CA}\cdot [\mathrm{Mg}(\mathrm{OH})(H_{2}O{)}_{\chi -1}]{]}_{\left(\mathrm{surface}\right)}^{1-n}+H_{2}O$$
(6)$$\begin{array}{c} 5[CA\cdot [Mg(OH)(H_{2}O{)}_{X-1}]{]}_{\left(\mathrm{surface}\right)}^{1-n}+Mg^{2+}+SO_{4}^{2-}+5OH^{-}\to \\ 5Mg(OH{)}_{2}\cdot MgSO_{4}\cdot 7H_{2}O_{\left(\mathrm{nucleus}\right)}+(5x-2)H_{2}O+5CA^{n-} \end{array}$$

Under the condition of incorporating a 40% iron tailing powder admixture, the changes in MOS foam cement before and after modification are illustrated in Figure 2. The hydration products in the unmodified MOS cement test block, shown in Figure 2a, primarily consisted of flaky ${\mathrm{Mg}(\mathrm{OH})}_{2}$, which were disorderly stacked together. However, Figure 2b shows that after adding modifiers, a large number of 5·1·7 crystal phases were interlaced within the test block, increasing the compactness of the system. The micro-aggregate effect of the iron tailing powder also contributed to the densification of MOS foam cement during hardening, with these particles overlapping the needle-like structure of the 5·1·7 crystal phase. The combined crystal phase structure was coarser, enhancing the strength support of the sample. An appropriate amount of iron tailing powder particles stabilized the porous structure in the MOS foam cement slurry, thus improving its strength.

Figure 3 shows the XRD patterns of MOS foam cement under different conditions. It was observed that after adding the modifier, the foamed cement resulted in the formation of a 5·1·7 crystal phase, and the crystal diffraction peaks of ${\mathrm{Mg}(\mathrm{OH})}_{2}$ and MgO were significantly enhanced. This indicated that the modifier facilitated the hydration reaction within the MgO $\cdot \mathrm{MgS}O_{4}\cdot 7H_{2}O$ system, leading to the complete dissolution of $\mathrm{MgS}O_{4}\cdot 7H_{2}O$ particles and their participation in the hydration reaction. This process provided the necessary material for the early formation of complexes on the surface of MgO particles.

Additionally, when comparing the XRD spectrum of modified MOS foamed cement before and after the addition of iron tailing powder, no new phase was detected. However, the intensity of the SiO_2_ diffraction peak was significantly increased due to the iron tailing powder, which contained substantial amounts of quartzite and magnetite. This ferromagnetic material offered a foundation for the functional expansion of microwave absorption in doped iron tailing powder materials. At this stage, the consistency of the cement paste increased, significantly reducing the active MgO in unit mass. Consequently, the hydration degree of the early cement paste was insufficient, and the MgO diffraction peak was slightly higher.

### 3.2. Effect of Dry Density on Compressive Strength and Water Absorption of Samples

In order to prepare light-weight and high-strength MOS magnesium oxysulfate foam cement composites, it was necessary to study not only the influence of factors such as mix ratio, modifier type, and admixture, but more importantly, also to study how much dry density is suitable for practical applications.

This study aimed to prepare MOS foam cement with dry density grades of 600 kg/m^3^, 700 kg/m^3^, 800 kg/m^3^, and 900 kg/m^3^. This was achieved by adjusting the foaming agent dosage while maintaining a 40% iron tailing doping amount. The mechanical properties and water absorption of samples with different dry densities are presented in Table 4.

Mix proportion: The molar ratio of MgO/MgSO_4_/H_2_O is 7:1:20 (The amount of composite modifier is 4‰).

According to Table 4, it was observed that as the dry density of MOS foam cement increased, the compressive strength also increased, while the water absorption decreased.

Yi Longsheng et al. [16] utilized iron tailing powder as an admixture to prepare foamed cement. The optimum ratio was 55% iron tailings and 45% Portland cement with additional admixtures. The results indicated that at a dry density of 700 kg/m^3^, the 28 day compressive strength of the sample reached 1.65 MPa. However, MOS foam cement exhibited a compressive strength of 6.14 MPa under a similar dry density, with a water ab-sorption rate of 7.98%. At this density, the foam cement exhibited a large porous structure and high porosity, leading to thinner pore walls and increased connectivity between pores, thereby reducing the compressive strength of the sample. Additionally, the water absorption correlated with the number of connected pores in the material. A higher porosity increased the number of interconnected pores, resulting in a higher water absorption of porous materials.

Furthermore, as the dry density increased, the internal pore walls of MOS foam cement became thicker, and the amount of gel material providing support increased. Macroscopically, this resulted in excellent mechanical properties and a higher compressive strength. Subsequently, when the dry density reached 900 kg/m3, the compressive strength was 7.69 MPa and the water absorption rate was 5.49%. This was attributable to the decreasing size of the pore structure and the increasing thickness of the closed pore walls. In the water absorption experiment, the ability of water to pass through the pore wall decreased with an increasing dry density, thereby decreasing the water absorption. Additionally, the incorporation of iron tailings altered the pore structure and the microstructure of hydration products. Iron tailing particles were adsorbed in the pores of the crystal structure network in MOS foam cement. A higher dry density resulted in a more closed small pore structure. The combination of iron tailing powder particles and the 5·1·7 crystal nucleus in MOS cement became more compact, making the crystal accumulation denser and significantly enhancing the water resistance of the MOS foam cement.

### 3.3. Pore Size Analysis of Modified Magnesium Oxysulfate Foam Cement

To study the influence of different pore size distributions on the mechanical proper-ties of MOS foam cement, the amount of iron tailing powder was controlled at 40%. The dry density of the foam concrete was regulated by adjusting the quantity of foaming agent and stirring time. The cross-sectional images of the test samples are shown in Figure 4. Pore structure parameters were measured and analyzed using the Image-Pro-Plus soft-ware, as depicted in the pore size distribution map in Figure 5.

Figure 4 clearly shows the changes in pore size and pore wall thickness with varying dry densities in foamed cement. As the dry density increased, the number of large apertures exhibited a decreasing trend, while the thickness of the pore wall increased accordingly. This adjustment resulted in fewer connected pores and a gradual increase in the number of closed pores, enhancing the water resistance of the material and effectively reducing water absorption.

According to Li Lin et al. [36], the Image-Pro-Plus concrete pore structure image analysis method was employed to analyze the pore structure of MOS foam cement. The characteristic parameters of various three-dimensional pore structures were calculated. In this study, the pore size distribution of foamed cement samples with dry densities ranging from A06 to A09 grades was primarily between 10 μm and 200 μm. As shown in Figure 5, the proportion of pores in this size range was 60%, 73%, 84% and 89%, respectively. The pro-portion of small pores (<200 μm) increased with higher dry densities. However, at A06 and A07 dry densities, a small number of harmful macropores exceeding 1200 microns appeared. The presence of such macropores led to an uneven distribution of pore sizes, resulting in different pressures within large and small pores. However, it led to the thinning of the pore walls, increasing local stress and ultimately decreasing mechanical properties. Additionally, the inclusion of large amounts of iron tailing powder in magnesium oxysulfate foam cement not only reduces the cost of traditional magnesium oxysate foam cement but also contributes to an increased utilization of solid waste and environmental protection.

## 4. Conclusions

In conclusion, the study result indicated that the addition of a modifier enhanced the hydration reaction process of the MgO-MgSO_4_-H_2_O system, facilitating the crystallization of MOS foam and stabilizing the formation of the main strengthening phase (5·1·7 crystal phase). The incorporation of a large amount of iron tailing powder enhanced the growth and coarsening of the MOS crystal phase. This resulted in a more compact and strengthened overall structure, a more uniform pore size distribution, and a stable crystal structure of the pore walls. The pores were predominantly circular closed pores. At a dry density of A08 grade and 40% content, the compressive strength of the sample reached 6.83 MPa, achieving a high-cost performance. There might be even better properties for magnesium oxysulfate foam cement, and we will continue to investigate.

## Figures and Tables

**Figure 1 materials-17-05907-f001:**
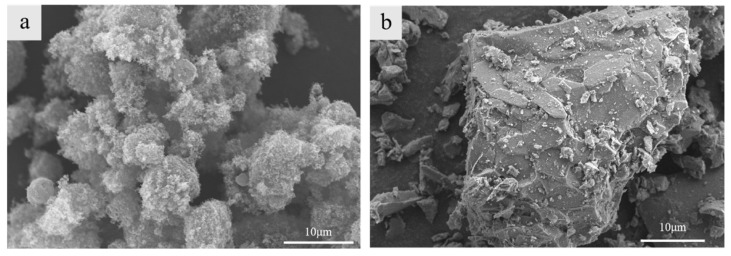
SEM photos of light-burned magnesia and iron tailings: (**a**) light-burned magnesia; (**b**) iron tailing powder.

**Figure 2 materials-17-05907-f002:**
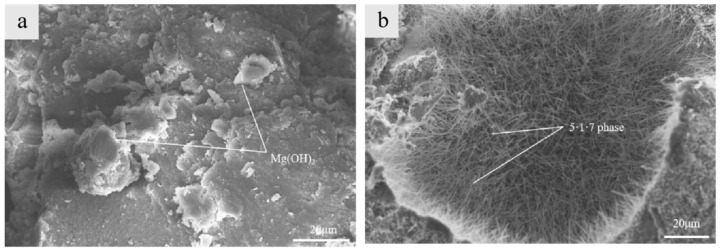
SEM images of magnesium oxysulfate foam cement before and after modification. (**a**) Before modification; (**b**) after modification.

**Figure 3 materials-17-05907-f003:**
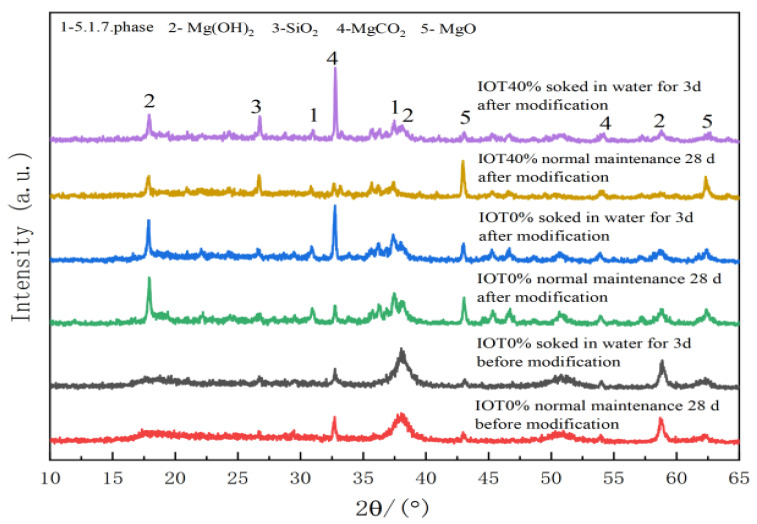
XRD spectra of magnesium oxysulfate foam cement before and after modification under different curing conditions.

**Figure 4 materials-17-05907-f004:**
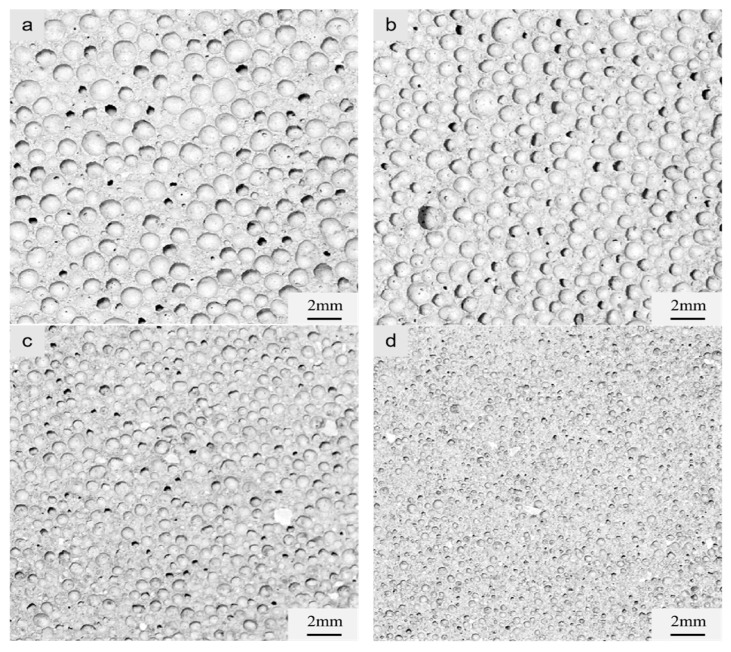
Cross-sectional images of magnesium oxysulfate foam cement samples with different dry densities: (**a**) A06 grade, (**b**) A07 grade, (**c**) A08 grade, and (**d**) A09 grade.

**Figure 5 materials-17-05907-f005:**
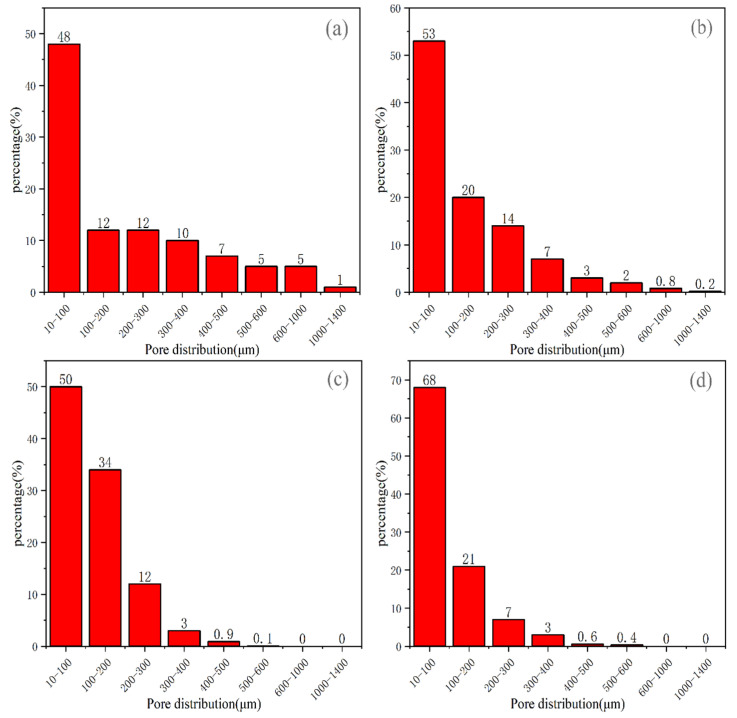
Pore size distribution of modified foam magnesium oxysulfate concrete at different dry densities: (**a**) A06 grade, (**b**) A07 grade, (**c**) A08 grade, and (**d**) A09 grade.

**Table 1 materials-17-05907-t001:** Chemical composition of light-burned magnesia raw materials.

Composition	MgO	SiO_2_	CaO	Fe_2_O_3_	Al_2_O_3_	Ignition Loss
Content (wt%)	86.41	5.79	1.56	0.32	0.64	5.28

**Table 2 materials-17-05907-t002:** Chemical composition of iron tailings.

Composition	SiO_2_	Fe_2_O_3_	Al_2_O_3_	CaO	MgO	K_2_O	BaO
Content (wt%)	37.9	38.7	8.29	5.81	2.99	1.95	1.48

**Table 3 materials-17-05907-t003:** Basic physical properties of magnesium oxysulfate foamed cement with different contents of iron tailings.

Samples	Iron Tailing Powder (wt%)	Compressive Strength (MPa)	Water Absorption (%)	Dry Density (kg/m^3^)
1	10	4.45	9.17	800
2	20	5.43	8.42	800
3	30	6.36	8.29	800
4	40	6.82	7.21	800
5	50	5.57	8.62	800
6	60	4.28	9.46	800

**Table 4 materials-17-05907-t004:** Basic physical properties of magnesium oxysulfate foam cement at different dry densities.

Samples	Dry Density (kg/m^3^)	Compressive Strength (MPa)	Water Absorption (%)
1	600	5.27	9.40
2	700	6.14	7.89
3	800	6.79	7.13
4	900	7.69	5.49

## Data Availability

The original contributions presented in this study are included in the article. Further inquiries can be directed to corresponding authors.

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
