# Peer review of "Modified Magnesium Oxysulfate Foam Cement Doped with Iron Tailings"

_materials, 2024, doi:10.3390/ma17235907_

Round 1
Reviewer 1 Report
Comments and Suggestions for Authors
Reviewer comments:
Modified magnesium oxysulfate foam cement doped with iron tailings
Reviewer comments:
The presented paper demonstrates the experimental results of foamed composite with the addition of iron tailings. The presented results may be interesting at the local level; however, they obviously do not satisfy the international level.
The novelty and the significance of the study were not disclosed. The presentation and interpretation of the results do not sound scientific. There are a lot of statements without proof. In my opinion, no scientifically interesting data is provided in the manuscript. Compressive strength and water absorption measurements, optical images, and XRD patterns did not reveal the significance of the results. Moreover, the authors did not provide a cost analysis; however, they concluded about “high-cost performance”.
Introduction
The authors cited around 30 papers, but they did not disclose what has already been done in the field of MOS foam cement production. It is unclear where the gap is and what the study's novelty is. The introduction should be expanded to describe what has already been done.
Methodology
Line 80 – foaming agent and foam stabilizer are not indicated and characterized.
Line 87-89. – It would be useful if the procedure of bulk density and water absorption were described in more detail.
Line 93 - the sample’s preparation procedure for XRD measurement should be mentioned.
Overall, the methodology of sample preparation remains unclear. I suggest the authors provide a graphical illustration of the process, including the main points, such as mixing time incorporation sequence.
Results
It was unclear why the dry density of the composite was selected at first 800kg/m3 , but later authors studied 600, 700 and 900kg/m3?
Line 126 – FIG.3 repeats the results presented in Table 3; therefore, one of them should be deleted.
Line 135 – what does CA mean? If you write the formulas, all components of the reactions should be clearly indicated.
Line 167 – FIG.5 is not correct and unclear – notations should be revised.
Line 176 – the judgment about the hydration degree of the system from the intensity of XRD peaks is incorrect, as the intensity indicates the orientation of the crystalline phase. Do you prepare samples properly?
Line 196-199 – text is more appropriate to the introduction. What is the purpose of this text?
Line 203-205 – your statements about pore connectivity should be proven; otherwise, they are just assumptions.
Line 207 – the statement about the pore thickness should be proven.
Line 2016 – what is the difference between voids and pores? At least, the size range should be provided.
Line 232 – Fig. 7 provides no useful information about the pore wall thickness. No measurements are presented. Where are the pores and voids in Fig.7?
Line 234 – how does Fig.7 explain the difference in water absorption between samples?
Conclusions
Line 257-258 – What do you mean by saying, “...a large amount of iron tailings powder enhanced the growth and coarsening of the MOS crystal phase.” ? Where is the MOS crystal phase?
Line 259 – what do you mean by saying “...a stable crystal structure of the pore walls”? Can you provide a cross-section of the pore (alone) wall?
Line 261 – where is your cost-saving analysis?
Reviewer 2 Report
Comments and Suggestions for Authors
It is an interesting study that investigated the influence of addition of iron tailings powder on the crystallization characteristics, pore structure, compressive strength, and water absorption of modified magnesium oxysulfate (MOS) foam cement with different dry densities. This study focus on preparing MOS foam cement with dry density grades of 600 kg/m3, 700 kg/m3, 800 kg/m3 and 900 kg/m3. The iron tailings powder content ranged from 10% to 60%.
I have just few questions and some advice for the authors:
1. Hyphen mistake in the following lines: line 10 (comprehensive), line 12 (tailings), line 37 (advantages), line 181 (factors), line 205 (increased), line 212 (decreasing), line 214 (decreased).
2. Abstract, line 13: water resistance – Did you mean water absorption? Considering experimental part of manuscript, I have not seen water impermeability test results.
3. Which type of cement did you use in the experiment? Maybe you should add this information in Chapter 2.1
4. Did you examine the pozzolanic reaction of iron tailings powder? Does iron tailings powder have this fines used in the experiment? Or you needed to grind it and sieve it before use for the production of MOS cement paste.
The paper is well organized, with good structure and I do not have further questions or suggestion.
Reviewer 3 Report
Comments and Suggestions for Authors
In the current manuscript, the authors examined the effects of high concentrations of iron tailings powder on the crystallization characteristics, pore structure, compressive strength, and water resistance of modified magnesium oxysulfate (MOS) foam cement with different dry densities. Furthermore, employing chemical foaming technology, the study characterized and analyzed the microstructure of modified MOS foam cement hydration products through scanning electron microscopy (SEM) and X-ray diffraction analysis (XRD). The results indicated that the addition of an acidic modifier effectively facilitated the hydration reaction in the MgO-MgSO4-H2O system, enhancing the micro-crystallization characteristics of MOS foam cement. The internal pores were uniformly round with a dense crystal structure within the pore walls. The com-pressive strength of the material with 40% dry density A08 grade iron tailings powder reached 6.83 MPa, and the lowest water absorption was 5.32% at a dry density of A09.
Generally, the paper is well written and structured. The research community would be interested in this work. However, before any decision is made, the following comments need to be addressed properly:
- If possible, please add some illustrative description from the production process of the magnesium oxide in the current investigation.
- What is the main advantage of “Magnesium oxysulfate cement (MOS)” utilized in magnesia cementitious material? Why is such a type of porous material needed in industry?
- If the “chemical composition” of light burned magnesia material is changed, what can be expected from the new material? Please add new discussion related to what effect can be expected due to the change in the composition of the considered porous materials.
- What is the optimum wt% of Iron tailing powder for the best performance of magnesium oxysulfate foamed cement in general?
- The introduction section lacks the sense of application of cement material and its different types in real world applications. It is highly recommended to enrich this section by discussing further works related to the mechanical properties of concrete substrates using different methodologies. The following articles in the literature are suggested to be included and discussed:
- https://doi.org/10.22190/FUME221215005H
- DOI: 10.22055/jacm.2020.32921.2101
- More discussions on the results are needed to present the major outcomes of the present study.
- Conclusions should include limitations of the work done (if the authors are aware of any) and some clear directions for the future work.
Reviewer 4 Report
Comments and Suggestions for Authors
Dear Authors
paper is quite complete. Minor corrections are suggested
In introduction, is there any statistics about rate of waste production per year on regional base, for mineral sector or for the other civil and industrial sectors?
Inn ch.2 iron mine tailings are originated from what kind of ore dressing technology and raw minerals?
How can be classified addives and stabilizers to tailings? Is there any comparison with cementitious products?
In fig 3 - 6 correct “Strength”.
Is there any image of specimen and testing apparatus for geomechanical testing? How deformability can be detected?
In Conclusions, is there any applicability of the mixed product? Is is stabilized for long term?
Round 2
Reviewer 1 Report
Comments and Suggestions for Authors
The authors considered my comments.
Reviewer 3 Report
Comments and Suggestions for Authors
The authors have suitably revised the paper. It is recommended for publishing as it is.